# GENERATIVE ADVERSARIAL NETWORKS USING ADAPTIVE CONVOLUTION

## ABSTRACT

Most existing GANs architectures that generate images use transposed convolution or resize-convolution as their upsampling algorithm from lower to higher resolution feature maps in the generator. We argue that this kind of fixed operation is problematic for GANs to model objects that have very different visual appearances. We propose a novel adaptive convolution method that learns the upsampling algorithm based on the local context at each location to address this problem. We modify a baseline GANs architecture by replacing normal convolutions with adaptive convolutions in the generator. Experiments on CIFAR-10 dataset show that our modified models improve the baseline model by a large margin. Furthermore, our models achieve state-of-the-art performance on CIFAR-10 and STL-10 datasets in the unsupervised setting.

## 1 INTRODUCTION

Generative Adversarial Networks (Goodfellow et al., 2014) (GANs) are an unsupervised learning method that is able to generate realistic looking images from noise. GANs employs a minimax game where a generator network learns to generate synthesized data from random noise and in conjunction, a discriminator network learns to distinguish between real and generated data. Theoretically, the training processes of the two networks intertwine and iterate until both networks reach a Nash equilibrium where real and synthesized data are indistinguishable.

However, in practice, GANs are notoriously hard to train. For the learning of the generator to happen effectively, hyper-parameters, as well as the architectures of the generator and discriminator, must be chosen carefully. Nevertheless, on datasets with visually similar images, such as bedroom scenes (Yu et al., 2015) and faces (Chen et al., 2016), GANs have produced good results (Radford et al., 2015). This success, however, does not translate to datasets that have diverse visual categories. GANs models trained on ImageNet (Russakovsky et al., 2015) were only able to learn basic image statistics and some shapes, but they did not learn any object (Salimans et al., 2016). Recent works address this problem by incorporating additional high-level information to guide the generator, such as training the discriminator in a semi-supervised manner (Salimans et al., 2016), adding a second training objective to direct the generator toward similar samples from the training set (Warde-Farley & Bengio, 2016) or using artificial class labels by clustering in the representation space (Grinblat et al., 2017).

We take a different approach to tackle this problem. We hypothesize that the rigidity of the normal convolution operator is partially responsible for the difficulty of GANs to learn on diverse visual datasets. Most GANs generators upsample low resolution feature maps toward higher resolution using fixed convolutions (note that a transposed convolution is equivalent to a convolution) or resize-convolution (Odena et al., 2016a). Such operations are unintuitive, because, pixel locations have different local contexts and belong to diverse object categories. Consequently, different pixel locations should have different upsampling schemes. This shortcoming of normal convolution is especially problematic in the early upsampling layers where higher level information usually associates with the object shapes and the context of images.

We propose the use of a novel adaptive convolution (Niklaus et al., 2017) architecture, called Adaptive Convolution Block, that replaces normal convolutions to address the aforementioned shortcoming of GANs generators. Instead of learning a fixed convolution for the upsampling of all pixels

from the lower to the higher resolution feature map, an AdaConvBlock learns to generate the convolution weights and biases of the upsampling operation adaptively based on the local feature map at each pixel location. AdaConvBlock helps the generator to learn to generate upsampling algorithms that take into account the local context. We believe that this kind of adaptive convolution is more intuitive and more akin to the process when a human draws something: the same thought process is used in the whole drawing but the style of each stroke should vary and depend on the local context around each pixel position.

We conduct experiments to compare our adaptive convolution method to normal convolution in the unsupervised setting. We progressively replace all convolutions of a GANs generator with AdaConvBlocks from the lowest resolution to the highest resolution. Experiments on CIFAR-10 dataset show that the modified adaptive convolution models have superior qualitative and quantitative performance over the baseline architecture and just replacing convolution of the upsampling from the lowest resolution feature map with adaptive convolution can have significant impacts on the baseline model. Furthermore, our best models achieve state-of-the-art unsupervised performance on both CIFAR-10 and STL-10 datasets. Our code and models will be released.

## 2 BACKGROUND

### 2.1 GENERATIVE ADVERSARIAL NETWORKS

Generative Adversarial Networks (Goodfellow et al., 2014) is a framework where a *generator* G that tries to mimic real data of a target distribution is pitted against a *discriminator* D that tries to distinguish the generated samples from the target distribution. G is trained to increase the chance that generated samples are classified as real data while D is trained to minimize it. The training processes of D and G alternate and can be formulated as a minimax game:

$$\min_G \max_D V(D,G) = \mathbb{E}_{x \sim q_{data}(x)}[\log D(x)] + \mathbb{E}_{z \sim p_x(z)}[\log\left(1 - D(G(z))\right)] \tag{1}$$

where $q_{data}(x)$ is the real data distribution on $\mathbb{R}^n$, $p_x(z)$ is a commonly used distribution such as $\mathcal{N}(0, I)$, $z \in \mathbb{R}^m$ is a random noise variable drawn from $p_x(z)$, $G : \mathbb{R}^m \rightarrow \mathbb{R}^n$ is a generator function that maps the random noise variable to the real data space, $D : \mathbb{R}^n \rightarrow [0,1]$ is a function that outputs the probability of a data point in $\mathbb{R}^n$ belonging to the target real data distribution. The training process of GANs takes turns to update the discriminator for a number of times before updating the generator once. Ideally, the discriminator should be trained to convergence before updating the generator. However, this is computationally infeasible and causes the $D$ to overfit on datasets with finite data.

In the framework of GANs, $G$ and $D$ can be any differentiable functions. For image generation, they are usually formulated as convolutional neural networks. The generator $G$ usually consist of a fully connected layer to project the random variable to a small 3D volume followed by upsampling layers using convolutions that progressively refine the volume to have the desired spatial dimensions while the discriminator $D$ is usually constructed as the reverse of the generator, using strided convolutions to downsample the feature maps.

### 2.2 DIFFICULTIES IN TRAINING GANS

The difficulties of training GANs is well known. For example, the balance between the strength of the generator and that of the discriminator is essential for successful training. If $D$ is too strong, $\log\left(1 - D(G(z))\right)$ will be close to zero and there would be almost no gradient from where $G$ could learn to generate data. On the other hand, if $D$ is too weak, it will not be able to provide a good feedback for $G$ to improve. Another prominent issue is mode collapse where $G$ learns to map the majority of the random distribution $p_x(z)$ to a few regions in the real data space, resulting in near duplicate images. Overall, the training of GANs is unstable and very sensitive to hyper-parameters.

Several works have been done to address the difficulties of training GANs. WGAN (Arjovsky et al., 2017) pioneered the conditioning of $D$ to be a Lipschitz function by using weight clipping. Danihelka et al. (2017) proposed an improved version, called WGAN-GP, that enforces this conditioning by penalizing the gradient of $D$ on the set of straight lines between real and generated samples. Spectral Normalization GANs (Miyato et al., 2017) is one of the most recent works in this category.

Spectral Normalization controls the Lipschitz constant of the discriminator by dividing the weight matrix of each layer by its own spectral norm, which is equal to its largest singular value. The largest singular value of a matrix can be efficiently computed by an approximation algorithm (Miyato et al., 2017). Spectral Normalization has been shown to make GANs robust to hyper-parameters choices without incurring any significant overhead. For these reasons, we use spectral normalization to train our architectures.

# 3 ADAPTIVE CONVOLUTION BLOCK FOR GENERATIVE ADVERSARIAL NETWORKS

Although a lot of progress has been made in improving the training procedure, datasets with visually diverse images still pose a challenge for GANs. GANs fail to produce good-looking samples even on low dimension datasets like CIFAR-10 and STL-10. In this paper, we propose a novel Adaptive Convolution Block (AdaConvBlock) as a replacement for normal convolutions in GANs generator to tackle this issue. The AdaConvBlock can be thought of as a way to increase the capacity of the generator, making it easier to learn the sophisticated multimodal distributions underlying the visually diverse datasets. The details of our network architectures that use AdaConvBlocks are shown in section 5.2. Note that the kind of AdaConvBlock we describe in this paper only replace a normal convolution. In the case of a transposed convolution, an Adaptive Transposed Convolution Block can be implemented by simply rearranging the input volume (in the same way when converting a transposed convolution to a normal convolution) first, then apply an AdaConvBlock to the rearranged volume. Due to implementation difficulties of this rearrangement operation in Tensorflow (Abadi et al.), the deep learning framework we use, we only experiment with Adaptive Convolution Blocks in this paper.

## 3.1 ADAPTIVE CONVOLUTION BLOCK

Consider a convolution operation with filter size $K_{filter} \times K_{filter}$ and number of output channels $C_{out}$ on an feature map that has $C_{in}$ input channels. At every location, a convolution requires a weight matrix $W$ of size $K_{filter} \times K_{filter} \times C_{in} \times C_{out}$ as the filters to be convolved with the local feature map that has spatial dimension of size $K_{filter} \times K_{filter}$ followed by adding a bias matrix $b$ of size $C_{out}$ to each channels of the previous convolution.

For a normal convolution, all spatial locations in the input feature map will have the same weight $W_{normal}$ and bias $b_{normal}$. The shared weight and bias matrices serve as the learned variables of a normal convolution.

For an adaptive convolution, however, all spatial locations do not share the same weight and bias variables. Rather, they share the variables that are used to generate the weight and bias for each pixel location based on the local information. For each pixel $(i, j)$, an adaptive convolution consists of two normal convolutions to regress the adaptive weight $W(i, j)$ and adaptive bias $b(i, j)$ at each location followed by the local convolution of $W(i, j)$ with the local feature map and the addition of $b(i, j)$ to the previous local convolution. In this case, the learnable variables of an adaptive convolution are the weights and bias matrices of the convolutions that are used to generate $W(i, j)$ and $b(i, j)$.

A naive AdaConvBlock learns four matrices $W_{w,w}$, $W_{w,b}$, $W_{b,w}$ and $W_{b,b}$ with the size of $K_{adaptive} \times K_{adaptive} \times C_{in} \times C_{adaptive}$, $C_{adaptive}$, $K_{adaptive} \times K_{adaptive} \times C_{out}$, $C_{out}$, in a serial order. $W_{w,w}$, $W_{w,b}$ are the weight and bias matrices of the convolution to regress the adaptive weight $W(i, j)$ for and $W_{b,w}$, $W_{b,b}$ are the weight and bias matrices of the convolution to regress the adaptive bias $b(i, j)$ for each pixel location. $K_{adaptive}$ is the filter size of the convolution (i.e. the size of the local window around the pixel location) in the input feature map to regress $W(i, j)$ and $b(i, j)$ from. $C_{adaptive} = K_{filter} \times K_{filter} \times C_{in} \times C_{out}$ is the number of output channels of the convolution to regress $W(i, j)$, which is equal to the size of the matrix $W_{normal}$ of a normal convolution. Note that $K_{adaptive}$ controls the amount of local information used in an AdaConvBlock and can be different from the regressed filter size $K_{filter}$. Denote $F_{in}$ as the input feature map, $F_{out}$ as the output feature map of a naive AdaConvBlock, the exact formulation of $F_{out}$ from $F_{in}$ is

described as below:

$$W_{adaptive} = ReLU(F_{in} * W_{w,w} + b_{w,b}) \tag{2}$$

$$b_{adaptive} = F_{in} * W_{b,w} + b_{b,b} \tag{3}$$

$$F_{out} = F_{in} *_{local} \overline{W}_{adaptive} + b_{adaptive} \tag{4}$$

where $W_{adaptive}$, $b_{adaptive}$ are the 3D volumes consisting of all adaptive convolution weights $W(i,j)$ and biases $b(i,j)$. Note that $W_{adaptive}$ contains all the weights $W(i,j)$ that have been flattened into vectors. $\overline{W}_{adaptive}$ denotes $W_{adaptive}$ after the adaptive weight matrices are reshaped back into the appropriate shape for convolution. $ReLU$ denotes the ReLU activation function. $*_{local}$ denotes the local convolution operator.

One drawback of a naive AdaConvBlock, however, is the extremely expensive operation of computing adaptive convolution weights from the input volume (i.e. $F_{in} * W_{w,w}$). The amount of memory and computation used by this operation grow proportionally to $K_{adaptive} \times K_{adaptive} \times C_{in} \times C_{adaptive} = K_{filter}^2 \times K_{adaptive}^2 \times C_{in}^2 \times C_{out}$. We use depthwise separable convolution (Sifre & Mallat, 2014) in place of normal convolution to reduce computation cost as well as memory usage of this operation. A depthwise separable convolution replaces a normal convolution with two convolutions: one convolution (called depthwise convolution) that acts separately on each channel followed immediately by a 1x1 convolution (called pointwise convolution) that mixes the output of the previous convolution into the number of output channels (Chollet, 2016). The first depthwise convolution has memory and computation costs proportional to $K_{adaptive}^2 \times C_{in} \times C_{depthwise}$ while the second pointwise convolution has memory and computation costs proportional to $C_{depthwise} \times K_{filter}^2 \times C_{in}^2 \times C_{out}$ with the depth multiplier $C_{depthwise}$ being the number of output channels for each input channel of the depthwise convolution. For the AdaConvBlocks in our architectures, cost of the pointwise convolution dominates cost of the depthwise convolution. By choosing $C_{depthwise} = 1$, this separation of one big convolution into two smaller convolutions cuts the amount of memory and computation cost of our models by roughly $K_{adaptive}^2$ times. Equation 2 is rewritten as:

$$W_{adaptive} = ReLU(F_{in} * W_{w,w,depthwise} * W_{w,w,pointwise} + b_{w,b}) \tag{5}$$

where $W_{w,w,depthwise}$ and $W_{w,w,pointwise}$ are the weight matrices of the depthwise and pointwise convolution that have size of $K_{adaptive}^2 \times C_{in}$ and $K_{filter}^2 \times C_{in}^2 \times C_{out}$, respectively.

Figure 1 illustrates the full structure of an AdaConvBlock. Note that we do not use Batch Normalization (Ioffe & Szegedy, 2015) in our AdaConvBlock.

## 3.2 DESIGN CHOICES OF AN ADACONVBLOCK

In this subsection, we discuss some design choices for the Adaptive Convolution Block.

First, both the adaptive convolution weights and biases do not have to be regressed necessarily from the input volume. Additional transformations can be applied to the input volume before regressing the weights and biases. We tried a few transformations and found them to cripple the performance of our network. For example, 3x3 dilated convolutions (Yu & Koltun, 2015) can be used to exponentially increase the receptive field to the regression of the weights and biases. The increase of receptive field can make object shapes more coherent. However, in practice, we found using multiple 3x3 dilated convolutions made training more unstable. The same effect can be achieved to by increasing $K_{adaptive}$ of the adaptive convolution without this downside. Another idea we tried was to add 1x1 convolutions before the regression to increase the nonlinearity of an AdaConvBlock. However, experiments showed that they were detrimental to the generator and hammered our model's performance.

Next, we considered the choice of activation functions and the lack of batch normalization in an AdaConvBlock. To regress both convolution weights and biases, we did not apply batch normalization as there were no reasons for the regressed weights and biases to follow any probability distribution. We applied a non-linearity after the convolution to regress the weights. Empirically, we found the ReLu activation made AdaConvBlock work better than other activation functions, including the identity activation (i.e. no activation). To regress the biases, we do not apply an activation function because doing so results in unwanted effects of limiting the output of an AdaConvBlock in a range.

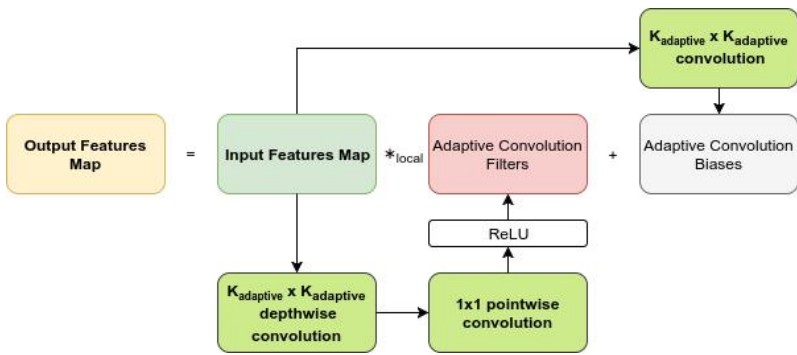

Figure 1: Diagram of an AdaConvBlock with local window size of $K_{adaptive}$.

Lastly, as described in section 3.1, to reduce the memory and computation cost, we used depthwise separable convolutions with a depth multiplier equal to one in place of normal convolutions while regressing adaptive convolution weights. The use of depthwise separable convolutions also had another benefit in that it made the memory and computation cost almost insensitive to the parameter $K_{adaptive}$ and allowed us to increase the receptive field to the regression at almost no cost. The choice of depth multiplier came from experiments. Empirically, we found increasing the depth multiplier not only increased the memory and computation cost but also slowed down the training progress. And overall, it did not improve our model's performance.

## 4 RELATED WORKS

There have been several works that seek to improve GANs performance on datasets that have high visual variability. Salimans et al. (2016) proposed a semi-supervised training procedure for GANs. Instead of learning to only distinguish real and fake samples, the discriminator also learns to classify which class the real data points belong to. Their method turns the discriminator into $K + 1$-way classifier, with $K$ classes of the real data and one class for the fake sample. Empirical results show that this kind of formulation surprisingly improves the quality of generated images. Based on the findings in this work, Warde-Farley & Bengio (2016) trained a denoising auto-encoder on the feature of the discriminator penultimate layer. For each generated sample, the squared difference between the discriminator feature and the reconstructed feature by the denoising auto-encoder for the penultimate layer is minimized. This additional training objective has the effect of guiding the generated samples toward regions in the feature space that correspond to higher probability configurations. The procedure is referred to by the authors as *denoising feature matching*. Grinblat et al. (2017) employed a simple but successful artificial class augmentation method for training GANs by dividing the samples using k-means clustering on the representation learned by the last hidden layer. Each cluster is treated as one artificial class. The networks are trained as an AC-GAN (Odena et al., 2016b) using the artificial class labels. The generator uses both the random noise variable $z$ and the artificial class label to generate fake samples while the discriminator tries to not only classify whether a sample is real or fake but to also construct the probability distribution over the artificial class labels. The discriminator starts with one cluster for the unsupervised case. Training progresses until a cluster is split into two when it has more samples than a threshold. Labels of the old cluster are then replaced with those of the new ones on the whole dataset. After this step, training is resumed with the new clusters.

The aforementioned methods have been successful to varifying degrees. However, the common theme among all of them is that they all try to make use of additional high level information, whether directly from the training labels or indirectly from the discriminator, to augment new training objectives that can direct the generator toward better sample generation. Our approach is different as we try to better the generator output by improving the architecture. Our method is complementary to these existing methods and a combination has the potential to yield better results.

Our method is inspired by the work of Niklaus et al. (2017) that applies adaptive convolution in video frame interpolation. The authors trained an encoder-decoder network to extract features on

Table 1: Architecture of the baseline generator. $M_g = 4$ for CIFAR-10 and $M_g = 6$ for STL-10.

| $z \in \mathbb{R}^{128} \sim \mathcal{N}(0, I)$ |
| :---: |
| dense $\rightarrow M_g \times M_g \times 128$ |
| neareast-neighbor 2x resize. 3x3, stride=1, 64 output channels conv. BatchNorm. ReLU |
| neareast-neighbor 2x resize. 3x3, stride=1, 32 output channels conv. BatchNorm. ReLU |
| neareast-neighbor 2x resize. 3x3, stride=1, 16 output channels conv. BatchNorm. ReLU |
| 3x3, stride=1, 3 output channels conv. Tanh |

two large image patches of the two video frames. The features are then fed into four subnetworks to estimate four 1D kernels that are then used for the interpolation. Although the base idea of using adaptive convolution is similar, there are differences between their work and ours that originates from differences in the problems. For the video interpolation task, they only have to regress a small number of outputs for each pixel location, while the size of our model, as well as outputs, grow cubically with the size of our base channels. This constraint makes the efficient use of memory more important in our work. Secondly, the filters of a video frame interpolation task are limited in the range $[0, 1]$ but that is not the case for our GANs convolution filters. Therefore, the design paradigms for the two architectures are different.

## 5 EXPERIMENTS

We perform experiments on CIFAR-10 (Krizhevsky, 2009) and STL-10 (Coates et al., 2011) datasets in a purely unsupervised setting. No labels or additional training objectives are used in the training process. Spectral Normalization (Miyato et al., 2017) is applied to the discriminator to stabilize training in all experiments. Zero padding is used for convolutions. All weights are initialized using a truncated normal distribution with mean zero and standard deviation of 0.02. Biases are initialized to zero. Following Miyato et al. (2017), we use Adam optimizer (Kingma & Ba, 2014) with $\alpha = 0.0002$, $\beta_1 = 0.5$, $beta2 = 0.999$ and batch size of 64 in all experiments. The number of discriminator updates per generator update is also fixed to one. Aligning with previous works, we compute the mean and standard deviations of the *Inception score* (Salimans et al., 2016) over 10 groups of 5000 randomly generated images. These two metrics are reported every 5000 training iterations and finally, the model with the highest mean score is selected for each architecture.

### 5.1 BASELINE

Our baseline architecture is based on the Spectral Norm GAN (Miyato et al., 2017) architecture. We replace all transposed convolution in the generator network with resize-convolution as an upsampling algorithm. The generator consists of six layers. The first layer is a Gaussian noise generator $\mathcal{N}(0, I)$ followed immediately by a fully connected layer to project the noise vector into a 3D volume that has spatial shape of a square with side of $M_g$ that depends on the dataset and depth of "base channels" equal to 512. We reduce the base channels of the baseline generator from 512 to 128. The reason is that our architectures using AdaConvBlocks can only fit into GPU memory with 128 base channels. Table 1 show the architecture of the baseline generator. Note that this baseline generator and the discriminator we use in this work are not balanced, which leads to a relatively low Inception score.

The discriminator network is kept unchanged from the work of Miyato et al. (2017). We use this discriminator for the baseline model as well as for all of our architectures.

### 5.2 OUR ARCHITECTURES

We progressively replace 3x3 convolutions from the third to the last layer of the baseline generator in Table 1 with AdaConvBlocks. Note that the 3x3 convolution in the last layer is not part of an upsampling step. However, in our experiments, we find that replacing this convolution also improves the performance of our model slightly. The AdaConvBlocks that replace normal convolutions must keep the filter size $K_{filter}$ and output channels $C_{out}$ intact, leaving the only one parameter left

Table 2: Architecture of AdaGAN. $M_g = 4$ for CIFAR-10 and $M_g = 6$ for STL-10. $K_{adaptive}$ for each AdaConvBlock are not specified.

| $z \in \mathbb{R}^{128} \sim \mathcal{N}(0, I)$ |
|---|
| dense $\rightarrow M_g \times M_g \times 128$ |
| neareast-neighbor 2x resize. $K_{filter} = 3, C_{out} = 64$ AdaConvBlock. BatchNorm. ReLU |
| neareast-neighbor 2x resize. $K_{filter} = 3, C_{out} = 32$ AdaConvBlock. BatchNorm. ReLU |
| neareast-neighbor 2x resize. $K_{filter} = 3, C_{out} = 16$ AdaConvBlock. BatchNorm. ReLU |
| $K_{filter} = 3, C_{out} = 3$ AdaConvBlock. Tanh |

that can vary is the size of the local window to regress the adaptive weights and biases $K_{adaptive}$. For a generator with base channel of 512, our architectures that use AdaConvBlocks do not fit into our GPU memory. The memory and computation cost of an AdaConvBlock grows cubically with the number of input channels $C_{in}$ and $C_{in}$ of the AdaConvBlocks, which are determined by the base channels. Therefore, we have to reduce the number of base channels from 512 to 128 for our architecture. Consequently, we have to reduce the base channels of our baseline generator as well.

We name our architectures based on the number of AdaConvBlocks used to replace normal convolution in the baseline model. For example, AdaGAN-1 is the model that has the 3x3 convolution in the third layer replaced with an AdaConvBlock, AdaGAN-2 is the model that has both convolutions in the third and the fourth layers replaced with AdaConvBlocks and AdaGAN-3 is the model that has all convolutions replaced except for the last layer. Additionally, we name AdaGAN as the model that has all 3x3 convolutions replaced with AdaConvBlocks. Table 2 shows the architecture of AdaGAN model. For AdaGAN-1, AdaGAN-2 and AdaGAN-3, their architectures can be derived easily from table 1 and table 2.

The choice of $K_{adaptive}$ is an important factor for the performance of our architectures. Ideally, $K_{adaptive}$ should be chosen separately for each layer. However, for simplicity, we fix $K_{adaptive}$ for all AdaConvBlocks in an architecture. We append $K_{adaptive} \times K_{adaptive}$ to the name of every architecture. For example, AdaGAN-1-3x3 is an AdaGAN-1 architecture that has $K_{adaptive}$ set to three, AdaGAN-5x5 is an AdaGAN architecture that has $K_{adaptive}$ set to five.

## 5.3 CIFAR-10

To show the effectiveness of AdaConvBlocks, we compare the performance of the baseline generator with our architectures on the CIFAR-10 dataset. We use $K_{adaptive} = 3$ for all AdaConvBlock in this experiment. We train all models for 200,000 iterations. Table 3 shows the Inception score of the baseline generator versus our architectures. Experimental results show that the Inception score increases with the number of AdaConvBlocks used in place of normal convolutions. Replacing the convolution in the first upsampling layer (layer three) with an AdaConvBlock has the highest impact, improving the mean Inception score from 6.55 to 7.30, a 0.75 points difference. The Ada-ConvBlock in this upsampling layer helps increase the generator capacity significantly, allowing the generator to counterbalance the discriminator strength and thus leads to much better training results. The benefits of AdaConvBlocks weaken gracefully in the subsequent layers and become negligible in the last layer. Our AdaGAN-3x3 architecture with 128 base channels beats Spectral Norm GAN (Miyato et al., 2017), which use normal convolutions, by a large margin, even though the latter uses a generator with 512 base channels and has arguably better balance. Therefore, the increases in Inception scores of our models compared to the baseline model cannot be attributed to the effect of balancing between the generator and discriminator alone and the flexibility induced by AdaConvBlocks must have played a major role. This confirms our hypothesis that using normal convolution in the upsampling layers limits the performance of the generator and adaptive convolution can be used to alleviate this problem.

To test the effects of $K_{adaptive}$, we additionally train another AdaGAN-5x5 model ($K_{adaptive} = 5$). This leads to a small increase in mean Inception score over the AdaGAN-3x3 model. Both of our AdaGAN models achieve state-of-the-art performance on CIFAR-10 dataset. Table 4, second column, shows the unsupervised Inception scores of our AdaGAN models compared to other methods on CIFAR-10. Figure 2 and 3 in appendix A show the samples generated by our AdaGAN models.

Table 3: Unsupervised Inception scores on CIFAR-10 of the baseline generator versus our architectures.

| Architecture | Inception score |
|---|---|
| Baseline | $6.55 \pm 0.08$ |
| AdaGAN-1-3x3 | $7.30 \pm 0.11$ |
| AdaGAN-2-3x3 | $7.74 \pm 0.06$ |
| AdaGAN-3-3x3 | $7.85 \pm 0.13$ |
| AdaGAN-3x3 | $\mathbf{7.96 \pm 0.08}$ |

Table 4: Unsupervised Inception scores on CIFAR-10 and STL-10

| Method | CIFAR-10 | STL-10 |
|---|---|---|
| Real Data (Warde-Farley & Bengio, 2016) | $11.24 \pm 0.12$ | $26.08 \pm 0.26$ |
| DFM (Warde-Farley & Bengio, 2016) | $7.72 \pm 0.13$ | $8.51 \pm 0.13$ |
| Spectral Norm GAN Miyato et al. (2017) | $7.42 \pm 0.08$ | $8.69 \pm 0.09$ |
| Splitting GAN ResNet-A Grinblat et al. (2017) | $7.90 \pm 0.09$ | $9.50 \pm 0.13$ |
| AdaGAN-3x3 | $7.96 \pm 0.08$ | $9.19 \pm 0.08$ |
| AdaGAN-5x5 | $\mathbf{8.06 \pm 0.12}$ | $9.67 \pm 0.10$ |
| AdaGAN-7x7 | | $\mathbf{9.89 \pm 0.20}$ |

## 5.4 STL-10

For STL-10 experiments, we train on the unlabeled set and downsample the images from $96 \times 96$ to $48 \times 48$, following Warde-Farley & Bengio (2016). As STL-10 has bigger image size than CIFAR-10, a larger $K_{adaptive}$ maybe helpful. Thus, we train an AdaGAN-7x7 model on this dataset as well. Our architectures converge much slower on STL-10 therefore we train our models for 400000 iterations. The two AdaGAN-5x5 and AdaGAN-7x7 models achieve state-of-the-art performance while the AdaGAN-3x3 model is just behind the work of Grinblat et al. (2017). Table 4, third column, shows the unsupervised Inception scores of our models against other methods. Figure 4, 5 and 6 in appendix A show the samples generated by our models.

## 6 DISCUSSION

We have demonstrated that using adaptive convolutions to replace normal convolutions in a GANs generator can improve the performance of a weak baseline model significantly on visually diverse datasets. Our AdaGAN models were able to beat other state-of-the-art methods without using any augmented training objectives. The samples generated by our models show that they seem to be able to learn the global context pretty well and be able to learn the rough shapes of the objects in most cases and the sample quality is quite reasonable on CIFAR-10 dataset. Furthermore, there are not much visible convolution artifacts in the generated images. The success of our models suggests that non-trivial performance improvement can be gained from modifying architectures for GANs.

The approach we take is different from other methods that try to inject high level information into the discriminator. These existing methods and AdaGAN can complement each other. More experiments need to be done, but we believe that our architectures can benefit from the augmented training objectives from existing methods.

Our method is not without a downside. Even though we used depthwise separable convolution to reduce the cost, the amount of memory and computation is still extremely high. More tricks could be applied to alleviate this issue. For example, in a similar manner to Niklaus et al. (2017) work, both the local convolutions and the convolution to regress the adaptive weights for the local convolutions in our AdaConvBlock can be approximated by separate 1-D convolutions. This can reduce the cost by more than 50%. Another idea is to exploit locality. We expect the adaptive convolution weights and biases of a pixel location to be quite similar to its neighbors and can be interpolated in a certain way. We will address this issue in our future work.

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

## A   SAMPLES GENERATED BY OUR MODELS ON CIFAR-10 AND STL-10 DATASETS

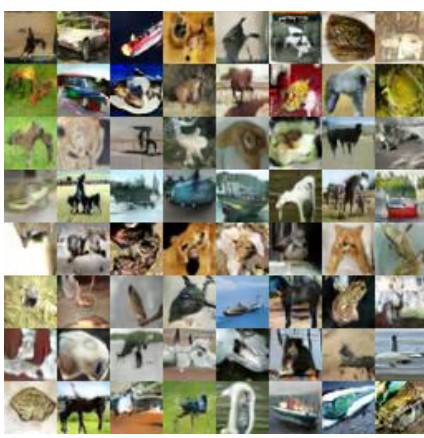

Figure 2: Samples generated by AdaGAN-3x3 on CIFAR-10 dataset

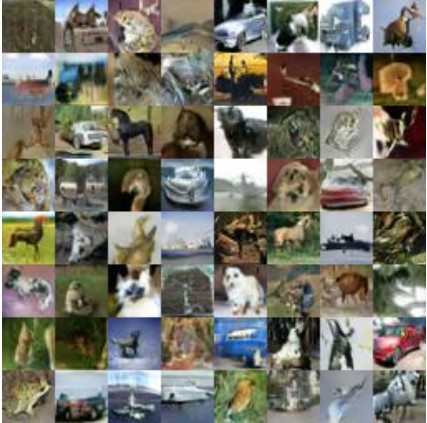

Figure 3: Samples generated by AdaGAN-5x5 on CIFAR-10 dataset

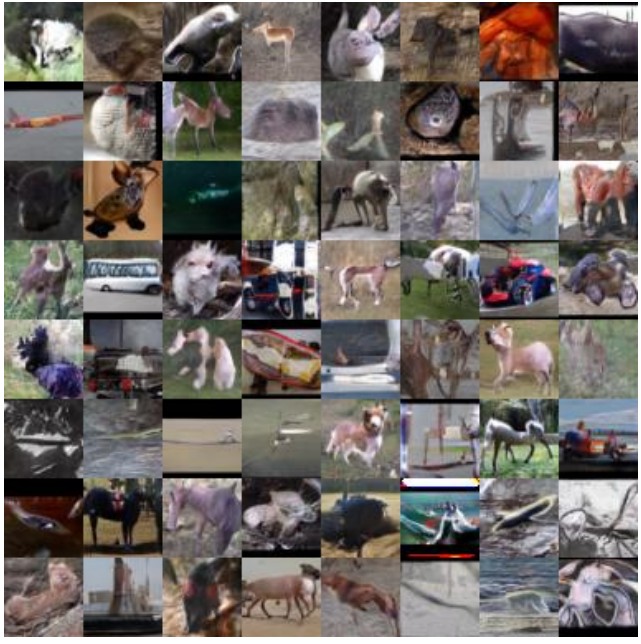

Figure 4: Samples generated by AdaGAN-3x3 on STL-10 dataset

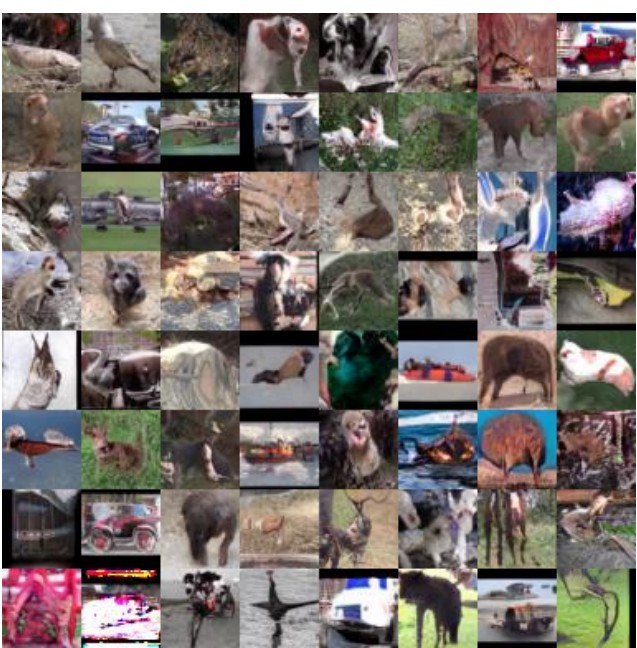

Figure 5: Samples generated by AdaGAN-5x5 on STL-10 dataset

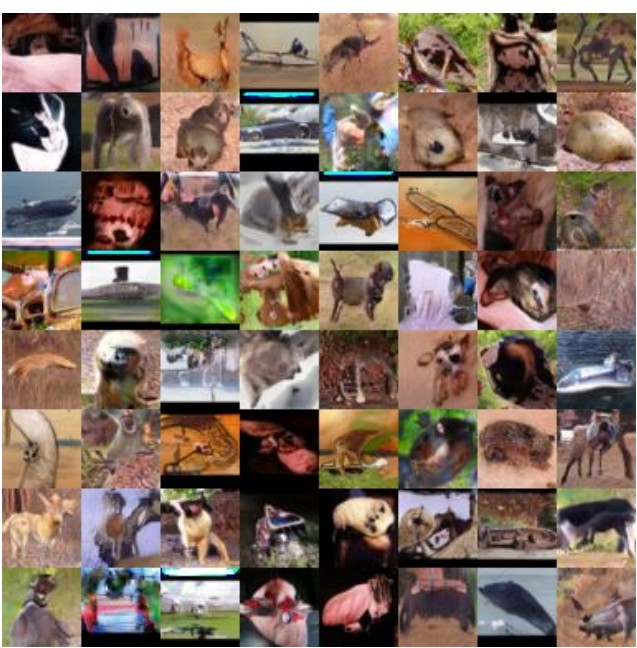

Figure 6: Samples generated by AdaGAN-7x7 on STL-10 dataset

