# OpenReview forum: "Generative Adversarial Networks using Adaptive Convolution"
_ICLR.cc/2018/Conference — Reject_

### Official Review · AnonReviewer3 · 2017-11-27

**Rating:** 4
**Confidence:** 5

**Review:**

The paper operates under the hypothesis that the rigidity of the convolution operator is responsible in part for the poor performance of GANs on diverse visual datasets. The authors propose to replace convolutions in the generator with an Adaptive Convolution Block, which learns to generate the convolution weights and biases of the upsampling operation adaptively for each pixel location. State-of-the-art Inception scores are presented for the CIFAR-10 and STL-10 datasets.

I think the idea of leveraging adaptive convolutions in decoder-based models is compelling, especially given its success in video frame interpolation, which makes me wonder why the authors chose to restrict themselves to GANs. Wouldn't the arguments used to justify replacing regular convolutions in the generator with adaptive convolution blocks apply equally well to any other decoder-based generative model, like a VAE, for instance?

I find the paper lacking on the evaluation front. The evaluation of GANs is still very much an open research problem, which means that making a compelling case for the effectiveness of a proposed method requires nuance and contextualization. The authors claim a state-of-the-art Inception score but fail to explain what argument this claim supports. This is important, because the Inception score is not a universal measure of GAN performance: it provides a specific view on the ability of a generator to cover human-defined modes in the data distribution, but it does not inform on intra-class mode coverage and is blind to things like the generator collapsing on one or a few template samples per class.

I am also surprised that the relationship with HyperNetworks [1] is not outlined, given that both papers leverage the idea of factoring network parameters through a second neural network.

Some additional comments:

- Figure 1 should be placed much earlier in the paper, preferably above Section 3. In its current state, the paper provides a lot of mathematical notation to digest without any visual support.
- "[...] a transposed convolution is equivalent to a convolution [...]": This is inaccurate. A convolution's backward pass is a transposed convolution and vice versa, but they are not equivalent (especially when non-unit strides are involved).
- "The difficulties of training GANs is well known": There is a grammatical error in this sentence.
- "If [the discriminator] is too strong, log(1 - D(G(z))) will be close to zero and there would be almost no gradient [...]": This is only true for the minimax GAN objective, which is almost never used in practice. The non-saturating GAN objective does not exhibit this issue, as [2] re-iterated recently.
- "Several works have been done [...]": There is a grammatical error here.
- The WGAN-GP citation is wrong (Danihelka et al. rather than Gulrajani et al.).

Overall, the paper's lack of sufficient convincing empirical support prevents me from recommending its acceptance.

References:

[1] Ha, D., Dai, A., and Le, Q. V. (2016). HyperNetworks. arXiv:1609.09106.
[2] Fedus, W., Rosca, M., Lakshminarayanan, B., Dai, A. M., Mohamed, S., and Goodfellow, I. (2017). Many Paths to Equilibrium: GANs Do Not Need to Decrease a Divergence At Every Step. arXiv:1710.08446.

---

### Official Review · AnonReviewer2 · 2017-11-30
**limited experiments**

**Rating:** 4
**Confidence:** 4

**Review:**

The paper proposes to use Adaptive Convolution (Niklaus 2017) in the context of GANs. A simple paper with: idea, motivation, experiments

Idea:
It proposes a block called AdaConvBlock that replaces a regular Convolution with two steps:
step 1: regress convolution weights per pixel location conditioned on the input
step 2: do the convolution using these regressed weights
Since local convolutions are generally expensive ops, it provides a few modifications to the size and shape of convolutions to make it efficient (like using depthwise)

Motivation:
- AdaConvBlock gives more local context per kernel weight, so that it can generate locally flexible objects / pixels in images

Motivation is hand-wavy, the claim would need good experiments.

Experiments:
- Experiments are very limited, only overfit to inception score.
- The experiments are not constructed to support the motivation / claim, but just to show that model performance improves.

Inception score experiments as the only experiments of a paper are woefully inadequate. The inception score is computed using a pre-trained imagenet model. It is not hard to overfit to.
The experiments need to support the motivation / claim better.
Ideally the experiments need to show:
- inception score improvements
- actual samples showing that this local context helped produced better local regions / shapes
- some kind of human evaluation supporting claims

The paper's novelty is also quite limited.

---

### Official Review · AnonReviewer1 · 2017-11-30
**Potentially interesting combination with little of its own novelty.**

**Rating:** 4
**Confidence:** 5

**Review:**

This manuscript proposes the use of "adaptive convolutions", previously proposed elsewhere, in GAN generators. The authors motivate this combination as allowing for better modeling of finer structure, conditioning the filter used for upsampling on the local neighbourhood beforehand.

While Inception scores were the only proposed metric available for a time, other metrics have now been introduced in the literature (AIS log likelihood bounds, MS-SSIM, FID) and reporting Inception scores (with all of their problems) falls short for this reviewer. Because this is just the combination of two existing ideas, a more detailed analysis is warranted. Not only is the quantitative analysis lacking but also absent is any qualitative analysis of what exactly these adaptive convolutions are learning, whether this additional modeling power is well used, etc.

---

### Decision · Program_Chairs · 2018-01-29
**ICLR 2018 Conference Acceptance Decision**

**Decision:**

Reject

**Comment:**

The paper proposes a GAN model with adaptive convolution kernels. The proposed idea is reasonable, but the novelty is somewhat minor and the experimental results are limited. More comprehensive experiments (e.g., other evaluation metrics) will strengthen the future revision of paper. No rebuttal was submitted.